# Cysteine–Cysteine Motif Chemokine Receptor 5 Expression in Letrozole-Induced Polycystic Ovary Syndrome Mice

**DOI:** 10.3390/ijms23010134

**Published:** 2021-12-23

**Authors:** Kok-Min Seow, Pin-Shiou Liu, Kuo-Hu Chen, Chien-Wei Chen, Luen-Kui Chen, Chi-Hong Ho, Jiann-Loung Hwang, Peng-Hui Wang, Chi-Chang Juan

**Affiliations:** 1Department of Obstetrics and Gynecology, Shin-Kong Wu Ho-Su Memorial Hospital, Taipei 111, Taiwan; m002249@ms.skh.org.tw (K.-M.S.); hwangskh@kimo.com (J.-L.H.); 2Department of Obstetrics and Gynecology, School of Medicine, National Yang Ming Chiao Tung University, Taipei 112, Taiwan; csu@ha.mc.ntu.edu.tw (C.-H.H.); phwang@vghtpe.gov.tw (P.-H.W.); 3Institute of Physiology, National Yang Ming Chiao Tung University, Taipei 112, Taiwan; emily110318@gmail.com (P.-S.L.); chho2@vghtpe.gov.tw (L.-K.C.); 4Department of Obstetrics and Gynecology, Taipei Tzu-Chi Hospital, The Buddhist Tzu-Chi Medical Foundation, Taipei 231, Taiwan; alexgfctw@mail.tcu.edu.tw; 5School of Medicine, Buddhist Tzu-Chi University, Hualien 970, Taiwan; 6College of Human Development and Health, National Taipei University of Nursing and Health Sciences, Taipei 112, Taiwan; gogozipper130@gmail.com; 7Department of Obstetrics and Gynecology, Taipei Veterans General Hospital, Taipei 112, Taiwan; 8Department of Obstetrics and Gynecology, Taipei Medical University, Taipei 110, Taiwan; 9Taipei IVF, Taipei 104, Taiwan; 10Institute of Clinical Medicine, National Yang-Ming University, Taipei 112, Taiwan; 11Female Cancer Foundation, Taipei 104, Taiwan; 12Department of Medical Research, China Medical University Hospital, Taichung 404, Taiwan; 13Department of Medical Research, Taipei Veterans General Hospital, Taipei 112, Taiwan

**Keywords:** polycystic ovary syndrome, letrozole, CCR5, CCL5

## Abstract

Polycystic ovary syndrome (PCOS), which affects 5–10% of women of reproductive age, is associated with reproductive and metabolic disorders, such as chronic anovulation, infertility, insulin resistance, and type 2 diabetes. However, the mechanism of PCOS is still unknown. Therefore, this study used a letrozole-exposed mouse model in which mice were orally fed letrozole for 20 weeks to investigate the effects of letrozole on the severity of reproductive and metabolic consequences and the expression of cysteine–cysteine motif chemokine receptor 5 (CCR5) in letrozole-induced PCOS mice. The letrozole-treated mice showed a disrupted estrous cycle and were arrested in the diestrus phase. Letrozole treatment also increased plasma testosterone levels, decreased estradiol levels, and caused multicystic follicle formation. Furthermore, histological analysis of the perigonadal white adipose tissue (pgWAT) showed no significant difference in the size and number of adipocytes between the letrozole-treated mice and the control group. Further, the letrozole-treated mice demonstrated glucose intolerance and insulin resistance during oral glucose and insulin tolerance testing. Additionally, the expression of CCR5 and cysteine-cysteine motif ligand 5 (CCL5) were significantly higher in the pgWAT of the letrozole-treated mice compared with the control group. CCR5 and CCL5 were also significantly correlated with the homeostasis model assessment of insulin resistance (HOMA-IR). Finally, the mechanisms of insulin resistance in PCOS may be caused by an increase in serine phosphorylation and a decrease in Akt phosphorylation.

## 1. Introduction

Polycystic ovary syndrome (PCOS) is a common endocrine metabolic disease that affects 5–10% of women of reproductive age [1,2]; it is associated with chronic anovulation, hyperandrogenism, and the development of multiple small subcapsular cystic follicles in the ovary (shown by ultrasonography) [3,4]. Furthermore, reproductive abnormalities, obesity, insulin resistance with compensatory hyperinsulinemia, dyslipidemia, and an increased risk of cardiovascular diseases and type 2 diabetes mellitus are frequently observed in women with PCOS [4,5]. Therefore, it is crucial to realize the mechanisms of PCOS to help decrease or treat PCOS complications. However, the etiology and pathogenesis of PCOS have not been completely investigated because of limited human studies due to ethical issues. Therefore, animal models are a valuable tool for the study of the pathogenesis, mechanisms, and long-term metabolism of PCOS, thereby identifying novel and more effective therapeutic strategies [6,7].

So far, there is no consensus on the best experimental animal model for the study of PCOS. According to [8], many methods have been used to induce a PCOS animal model, including exposure to androgen [9,10], estrogen [11], genetic modification, constant light [7], stress, and prenatal androgenization [12]. Androgen exposure is the most widely used method of inducing a PCOS animal model because the hyper-production of androgen in early life is thought to be the main cause of PCOS [10,13]. Excess androgen can also induce metabolic abnormalities, such as impaired glucose tolerance, and significant reproductive disturbances, including anovulation and ovarian cyst formation [13]. By injecting immature rats with a daily dose of DHEA (dehydroepiandrosterone), an animal model is created that exhibits an increased level of testosterone and is both anovulatory and acyclic, mimicking the typical features of hyperandrogenism in PCOS [9,13]. However, DHEA was reported to reduce body weight, serum glucose, and insulin and triglyceride levels in diet-induced obese male rats. This is contrary to the increased body mass index, hyperglycemia, and hyperinsulinemia seen in women with PCOS. Further, our previous study demonstrated that the DHEA treatment did not show insulin resistance in female rats even though the DHEA-treated rats had reproductive abnormalities that mimicked human PCOS symptoms. Therefore, we suggest that the DHEA-treated rats are not good animal models for studying metabolic abnormalities in PCOS [10]. Therefore, it is important to determine a better animal model for PCOS instead of the DHEA-treated rats.

Cysteine–cysteine (C-C) chemokine receptor type 5 (CCR5) is the most-studied receptor for the chemokine CC motif ligands (CCL) 3, CCL4, and CCL5 [14]. CCR5 was initially found as a co-receptor for the human immunodeficiency virus (HIV) infection of macrophages [15]. Additionally, recent evidence has suggested that CCR5 is associated with type 1 diabetes [16]. The CCR5 deletion polymorphism, CCR5delta32, is associated with a reduced risk of cardiovascular disease; CCR5 antagonism and gene deletion reduce atherosclerosis in animals [17]. Furthermore, CCR5 has been correlated with obesity, adipose tissue inflammation, and insulin resistance in both animal and human studies [18,19], and acute exercise may upregulate CCR5 expression in the skeletal muscles of patients with PCOS [20]. A recent study also revealed that CCR5 is associated with the regeneration, angiogenesis, and immune response of nerves from four to seven days after injury [21]. Further, the autosomal recessive deficiencies of CCR5 underlie resistance to HIV-1 [22]. Maraviroc, a CCR5 antagonist, reduces liver fibrosis and injury, chronic liver disease, and tumor burden in mice fed a hepatocarcinogenic diet [23]. Another recent study demonstrated that CCR5 expression in adipose tissue and peripheral blood mononuclear cells was significantly higher in women with PCOS compared with women in the control group [24]. CCR5 was also upregulated in the THP-1 cells after chronic exposure to testosterone [24]. However, the relationship between CCR5 and PCOS is still inconclusive.

Recently, letrozole, a non-steroidal inhibitor of P450 aromatase, successfully induced PCOS in rats, as shown in several studies [25,26,27]. Continuous administration of letrozole at 200 μg/d starting before puberty induces reproductive abnormalities and metabolic disturbances in female rats, mimicking the symptoms of women with PCOS [26]. In addition, these rats exhibited increased body weight and inguinal fat accumulation, anovulation, larger ovaries with follicular atresia and multiple cysts, endogenous hyperandrogenemia, and lower estrogen levels [26]. Furthermore, letrozole-treated rats showed insulin resistance and enlarged adipocytes in perirenal and visceral fat depots, increased circulating levels of luteinizing hormone, decreased levels of follicle-stimulating hormone (FSH), and increased ovarian expression of Cyp17a1 mRNA [28]. These data indicated that letrozole-treated rats might be good animal models for studying the mechanisms and pathogenesis of PCOS.

Conclusively, this study used a letrozole-exposed mouse model in which mice were orally fed letrozole for 20 weeks to investigate the effects of letrozole on the severity of reproductive and metabolic consequences. In this model, CCR5 expression in letrozole-induced PCOS mice may also be investigated.

## 2. Results

### 2.1. Abnormalities of the Estrous Cycle in the Letrozole-Treated Mice

After 4 weeks of orally consuming letrozole (35 mg letrozole/kg), the letrozole-treated mice began exhibiting irregular estrous cycles, and their vaginal smears showed keratinized squamous epithelial cells; leukocytes were the predominant cell type, indicating the diestrus phase. The control mice (fed only commercial chow diet) showed normal estrous cycles with vaginal cell morphology variations progressing from proestrus, estrus, metestrus, and diestrus phases in 4–5 d (Figure 1A). This indicated that the cycle of the letrozole-treated mice was disrupted and arrested in the diestrus phase (Figure 1B).

### 2.2. Plasma Sex Steroid Levels in the Letrozole-Treated Mice

Plasma testosterone levels significantly increased in the letrozole-treated mice compared with the control mice (*p* < 0.01) (Figure 2A), indicating the presence of hyperandrogenism. In contrast, plasma estradiol levels were significantly lower in the letrozole-treated mice compared with the control mice (*p* < 0.05) (Figure 2B).

### 2.3. Ovarian Morphology and Weight in the Letrozole-Treated Mice

There were visible cysts on the surface of the ovaries in the letrozole-treated mice after 20 weeks of oral feeding with letrozole (Figure 3A). Additionally, the weight of the ovaries in the letrozole-treated mice was significantly higher than in the control mice (*p* < 0.01) (Figure 3B). The control mice showed normal ovarian features during all stages of ovarian follicle development and post-ovulatory corpus luteum (CL) formation (Figure 3C). However, light microscopy showed an abnormal ovarian structure with an increased number of large and small cystic follicles in the letrozole-treated mice. Furthermore, no CL was observed in the ovaries of the letrozole-treated mice (Figure 3C).

### 2.4. Body Weight in the Letrozole-Treated Mice

After 20 weeks of treatment, the letrozole-treated mice were significantly heavier than the control mice (*p* < 0.01) (Figure 4A). The body weight gain was also higher in the letrozole-treated mice compared with the control mice (*p* < 0.01) (Figure 4B). However, the food intake (kcal) did not show a statistically significant difference between the letrozole-treated mice and the control mice (Figure 4C).

### 2.5. Adiposity in the Letrozole-Treated Mice

Histological analysis of the perigonadal white adipose tissue (pgWAT) showed no significant alteration in the number of adipocytes under H&E staining (Figure 5A,B). Further measurement of the size of the adipocytes using virtual microscope software showed that there was a trending increase in adipocyte size between the letrozole-treated and the control mice, but there was no statistically significant difference (B).

### 2.6. Plasma Lipid Profiles

Plasma triglyceride and total cholesterol levels were significantly higher in the letrozole-treated mice compared with the control mice (*p* < 0.01) (Figure 6A,B).

### 2.7. Glucose Homeostasis and Insulin Sensitivity Test

After 8 weeks of treatment with letrozole, the fasting plasma insulin levels were significantly higher in the letrozole-treated mice compared with the control mice; however, there was no significant alteration in the fasting glucose levels. After 16 weeks of oral letrozole feeding, there was no significant difference in the fasting insulin levels between the two groups, but the fasting glucose levels were significantly higher in the letrozole-treated mice than in the control mice (Figure 7A). Furthermore, the HOMA-IR scores were significantly higher in the letrozole-treated mice after 8 and 16 weeks of treatment with letrozole. HOMA-β scores showed a decreasing trend in the letrozole-treated mice after 16 weeks of oral feeding with letrozole compared with the control mice but did not show statistical significance (*p* < 0.1) (Figure 7B).

In the oral glucose tolerance test, the plasma glucose and insulin levels were significantly higher at 60 and 120 min in the letrozole-treated mice (Figure 8A). Furthermore, using the ΔAUC calculation, the letrozole-treated mice showed significantly higher levels than the control mice (*p* < 0.01) (Figure 8B).

### 2.8. CCR5 and CCL5 Expression in the Letrozole-Treated Mice

CCR5 and CCL5 expression in multiple organs or tissues of the letrozole-treated mice were assessed to understand the role of CCR5 in PCOS pathogenesis. Additionally, CCR5 and CCL5 expression in the pgWAT was significantly higher in the letrozole-treated mice compared with the control mice. However, CCR5 expression in the liver and skeletal muscles was significantly lower in the letrozole-treated mice than in the control mice (Figure 9A,B). CCL5 expression in the ovaries and the pgWAT of the letrozole-treated mice was significantly higher than in the control group, but no difference in expression was found in the liver and skeletal muscles.

### 2.9. Insulin Signal Transduction Pathway in Letrozole-Treated Mice

After 20 weeks of treatment with letrozole, a Western blotting analysis of the pgWAT was performed to investigate the mechanisms of insulin resistance in the insulin signal transduction pathways of the letrozole-treated mice (Figure 10A). Western blotting showed that Akt phosphorylation was significantly decreased (*p* < 0.01) (Figure 10B), and IRS-1-Ser307 phosphorylation was significantly increased (*p* < 0.05) (Figure 10C) in the letrozole-treated mice, but there was no significant difference in IRS-1-Tyr941 phosphorylation (Figure 10D).

### 2.10. CCR5 and CCL5 Correlation with Parameters

CCR5 is significantly positively correlated with testosterone (r2: 0.634; *p* = 0.005), body weight (r2: 0.735; *p* = 0.001), fasting insulin (r2: 0.476; *p* = 0.03), and HOMA-IR (r2: 0.473; *p* = 0.03) but not significantly correlated with fasting glucose (r2:0.136; *p* = 0.06). CCL5 did not correlate with testosterone, but significantly correlated with body weight (r2: 0.824; *p* = 0.0003), fasting glucose (r2:0.70; *p* = 0.002), fasting insulin (r2:0.477; *p* = 0.02), and HOMA-IR (r2: 0.664; *p* = 0.004) after 8 weeks of oral letrozole feeding.

## 3. Discussion

PCOS is characterized by multicystic ovaries, amenorrhea, glucose intolerance, insulin resistance, and cardiovascular disease [4]. The letrozole-treated mice showed significantly higher testosterone levels and lower estradiol levels compared with the control mice. Additionally, the letrozole-treated mice were more likely to develop glucose and insulin resistance, especially after OGTT. We successfully induced the animal model of PCOS-IR using the continuous oral feeding of letrozole for 20 weeks, similar to other studies using the continuous-release pellet implantation of letrozole. Furthermore, the letrozole-treated mice showed PCOS-like reproductive and metabolic phenotypes compared with the control group. Therefore, this model is suitable to be used as an animal model to investigate the mechanisms of insulin resistance in PCOS.

The body weight of the letrozole-treated mice was significantly increased compared to that of the control mice. Further, the body weight gain was also significantly increased in the letrozole mice even though their food intake was not significantly different compared with the control mice. Further investigation also found that the adipocytes number did not significantly differ between the letrozole-treated and the control mice. There was a trend of increase in adipocyte size between the letrozole-treated and the control mice, but the difference was not significant. These results indicate that letrozole and testosterone may induce abnormal adipose tissue distribution without changes in the adipocyte number and size. This is why obesity is more prevalent in women with PCOS than in the general population, and alterations in adiposity may be a risk factor for metabolic PCOS disorders.

The mechanism of insulin resistance in PCOS is still unclear. Several studies have demonstrated that PCOS manifests a post-binding defect in insulin signaling in the adipocytes and a decreased activity of PI3-kinase during muscle biopsies using euglycemic hyperinsulinemic clamps [29,30,31]. Human studies evaluating skeletal muscle and adipose tissue demonstrated that decreased tyrosine phosphorylation and increased serine phosphorylation on the insulin receptor substrates (IRS) 1/2 might be a mechanism of insulin resistance in PCOS [29,32]. In this study, we found similar results showing that IRS-1-Ser307 phosphorylation was significantly increased in the letrozole-treated mice even though IRS-1-Tyr941 phosphorylation did not show a significant difference when compared with the control mice. This animal study further proved that the post-binding defect in the insulin signaling in IRS-1/2 might be the primary mechanism of IR in PCOS.

CCR5 is a protein found on the surface of leukocytes and is also a chemokine receptor [14]. CCR5 is associated with type 2 diabetes, obesity, and insulin resistance in animal studies [16]. Our study also showed that CCR5 and CCL5 expression is significantly higher in the pgWAT of the letrozole-treated mice compared with the control group. Further, CCR5 is significantly associated with fasting insulin and HOMA-IR levels. A previous study by Kitade et al. [33] reported that CCR5 was upregulated in the WAT of genetically (ob/ob) and high-fat diet (HFD)-induced obese (DIO) mice, and the CCR5^-/-^ mice were protected from insulin resistance, indicating that CCR5 may play a role in insulin resistance. A recent study also found that CCR5 knockout significantly attenuated the glucose area under curve of OGTT and HOMA-IR in HFD-induced rats [34]. However, we could not provide any experimental evidence to establish the direct link between CCR5 and CCL5 and HOMA-IR in the letrozole-induced PCOS mice. This is a limitation of this study. A further study has been performed to investigate the mechanisms and relationship between CCR5 and CCL5 and insulin resistance. In addition, previous studies have presented that hyperandrogenism is an important factor in PCOS mechanisms. This study demonstrated that CCR5 is significantly associated with testosterone and body weight, indicating that CCR5 is associated with PCOS pathogenesis.

The role of CCL5 in the mechanism and pathogenesis of insulin resistance is still unknown. Shen et al. reported that CCL5 is involved in the development and maturation of ovarian follicles [35]. Furthermore, the elevation of CCL5 expression attenuated preantral follicle growth, survival, and estradiol secretion [35]. Further, CCL5 promoted follicular granulosa cell apoptosis and the inhibition of the PI3K/Akt pathway. Our study demonstrated that CCL5 expression in the ovaries and the pgWAT of the letrozole-treated mice is significantly higher than in the control group. Additionally, the expression of Akt was significantly decreased (*p* < 0.01) in the letrozole-treated mice, consistent with Shen’s study. This result indicated that CCL5 might be involved in the pathogenesis of insulin resistance in PCOS through the inhibition of Akt phosphorylation.

Recent guidelines recommended that an insulin-sensitizer be used as the main drug to improve insulin resistance and fertility in PCOS patients if the first-line treatment of lifestyle change with weight loss and physical activity fails. Metformin is the most common insulin-sensitizer used to improve the reproductive and metabolic abnormality in women with PCOS. Inositols and myo-inositols are other insulin-sensitizers that were found to have important effects on ovulation and metabolism in the treatment of PCOS [36,37]).

Conclusively, oral feeding of letrozole successfully induces PCOS-like animal models that exhibit reproductive and metabolic disturbances, mimicking the typical features of PCOS. CCR5 and CCL5 expressions were significantly higher in the pgWAT of the letrozole-treated mice compared with the control group. Furthermore, CCR5 and CCL5 were associated with the mechanisms of insulin resistance in PCOS through increased serine phosphorylation and inhibition of Akt phosphorylation.

## 4. Materials and Methods

### 4.1. Animal

Six-week-old C57BL/6 (wild-type) mice were purchased from the National Laboratory Animal Center, Taipei, Taiwan, and housed with four mice in one cage at a temperature of 20–22 °C. The mice were kept in a light-controlled room on an alternate 12 h light/12 h dark cycle (lights on, 0800). The mice were fed with a commercial chow diet (LabDiet 5001) and tap water ad libitum.

### 4.2. Study Procedure

After one week of acclimatization, the mice were randomly divided into two groups (*n* = 10 each). The control animals were fed with a laboratory rodent diet (protein, 28.507%; carbohydrate, 57.996%; and fat, 13.496) for 20 weeks. The letrozole-treated mice were fed with 37.5 mg per kg letrozole (Femara, Novartis Pharma AG, Basel, Switzerland), dissolved in a laboratory rodent diet for an equivalent time. Body weights were measured weekly from 21 d to the end of the experiment. At the end of the experiment, the mice in both groups were sacrificed by decapitation. The whole-body fat distributions of both groups of mice were determined. The procedure of the experiments is Figure 11.

### 4.3. Vaginal Smear

After oral feeding of letrozole for 4 weeks, the estrous cycle stages for both groups of mice were determined using microscopic analysis of the predominant cell type in daily vaginal smears for 10 d. Four estrous cycle stages were determined using the main cell types identified in the vaginal smears: proestrus, round nucleated epithelial cells; estrus, cornified squamous epithelial cells; metestrus, cornified squamous epithelial cells and leukocytes; and diestrus, nucleated epithelial cells and leukocytes. 

### 4.4. Blood Sampling and Biochemical Analysis

Blood sampling was conducted after overnight fasting. Blood samples for glucose and insulin measurements were collected by tail bleeding using a 1.5 mL heparin-coated polyethylene microcentrifuge tube on ice. In addition, trunk blood was collected from each mouse after decapitation. Plasma was separated by centrifugation and stored at −20 °C until assay. Plasma glucose was measured using a glucose analyzer (Model 23A, Yellow Springs Instrument Company, Yellow Springs, OH, USA), and plasma insulin was determined using a commercial ELISA kit (Mercodia AB, Uppsala, Sweden). Triglyceride and total cholesterol levels were measured using an enzymatic calorimetric kit (Diagnostic Systems GmbH, Holzheim, Germany). Testosterone and 17β-estradiol were measured using commercial ELISA kits (Cayman Chemical Company, Ann Arbor, MI, USA).

### 4.5. Oral Glucose Tolerance Test (OGTT)

After overnight fasting, zero-minute blood samples were taken from each mouse, and the mice were immediately given a glucose solution (concentration: 0.1 g/0.1 mL; 0.2 mL/100 g body weight) by gavage, and four more blood samples were collected at 30, 60, 90, and 120 min. Plasma insulin and glucose concentrations were determined as previously described [10]. Additionally, the area under curve (AUC) of glucose against time was calculated. The homeostasis model assessment of insulin resistance (HOMA-IR) was calculated as fasting insulin (mIU/L) × fasting glucose (mmol/L)/22.5. Finally, HOMA-β was determined as 20× fasting insulin (mIU/L)/fasting glucose (mmol/L) − 3.5.

### 4.6. Insulin Tolerance Test (ITT)

For ITT, the mice were subjected to overnight fasting and then IP injected with 0.75 U/kg of regular human insulin (Novo Nordisk, Clayton, NC, USA) without anesthesia. Then blood samples were collected before and after 15, 30, 60, 90, and 120 min insulin injections [19]. Whole-blood glucose levels were determined using a OneTouch glucose analyzer (LifeScan Inc., Milpitas, CA, USA).

### 4.7. Western Blotting

Tissues were lysed with a lysis buffer (0.5% Nonidet P-40, 1% Triton X-100, 10 mM Tris-base, 150 mm NaCl, 10% glycerol, 1 mM EDTA, 1 mM EGTA, and 1 mM phenylmethylsulfonylfluoride), and tissue lysates were made by sonication in the lysis buffer. Samples were resolved on 7.5% SDS-PAGE gels, and the contents of the gels were transferred onto polyvinylidene difluoride (PVDF) membranes. The membranes were pre-blotted in a skimmed milk buffer and immunoblotted with phosphorylated-IRβ and IRβ primary antibodies followed by secondary antibodies. In addition, horseradish peroxidase-conjugated secondary antibodies were used in conjunction with a chemiluminescence reagent.

### 4.8. RNA Extraction

Ovarian and periovarian adipose tissues were lysed, and the total RNA was extracted using a Tri Reagent Kit (Applied Biosystem, Waltham, MA, USA). The RNA concentration was determined by ultraviolet light absorption at 260 nm, and the integrity of the extracted total RNA was examined using 1% agarose gel electrophoresis. Finally, the RNA samples were incubated with RNase-free DNase I at 37 °C for 30 min, and then at 100 °C for 10 min to inactivate the DNase I.

### 4.9. RT-PCR Analysis of mRNA Levels

After digestion with DNase I, 2 µg of total RNA from each RNA sample was reverse-transcribed at 37 °C for 2 h using random primers and an RT reverse-transcriptase (Thermo Fisher, Inc. Waltham, MA, USA) to obtain the cDNA product. Two microliters of the cDNA product were reacted in a total volume of 50 μL PCR reaction solution and incubated at the following conditions: 1 cycle of 95 °C for 5 min; 35 cycles of 95 °C for 1 min, 55 °C for 1 min, and 72 °C for 1 min; and a final 20-min extension period at 72 °C. The probe was obtained from Thermo Fisher (Thermo Fisher, Inc., Waltham, MA, USA); the primers used were GAPDH (Mm99999915_g1), CCR5 (Mm01963251_s1), and CCL5 (Mm01302427_m1). Ten microliters of each studied gene and β-actin PCR products amplified from the same RT template solution were combined and electrophoresed on a 2% agarose gel and stained with ethidium bromide. Finally, the relative levels of mRNA expression to β-actin were detected under ultraviolet light and quantified.

### 4.10. Histological Analysis

Ovarian and perigonadal white adipose tissues were fixed with 4% paraformaldehyde, and the 5 μm-thick paraffin-embedded tissues were stained with H&E solution. The slides were photographed under a microscope and then scanned and analyzed using ImageScope virtual microscopy software (Aperio Technologies, Vista, CA, USA) [38] to measure the size of adipocytes and ovarian cysts. In addition, two researchers counted the number of follicles in the ovarian sections to ensure accuracy.

### 4.11. Statistical Analysis

The experiments were repeated at least four times. The results are expressed as means ± SD. Statistical significance was assessed using a one-way analysis of variance or the Student’s *t*-test. Correlations between CCR5, CCL5, and the parameters were performed using the Pearson correlation test. A *p*-value less than 0.05 was considered to be statistically significant. The analysis was conducted using Statistical Package for the Social Sciences v.26 (IBM Corp., Armonk, NY, USA).

## Figures and Tables

**Figure 1 ijms-23-00134-f001:**
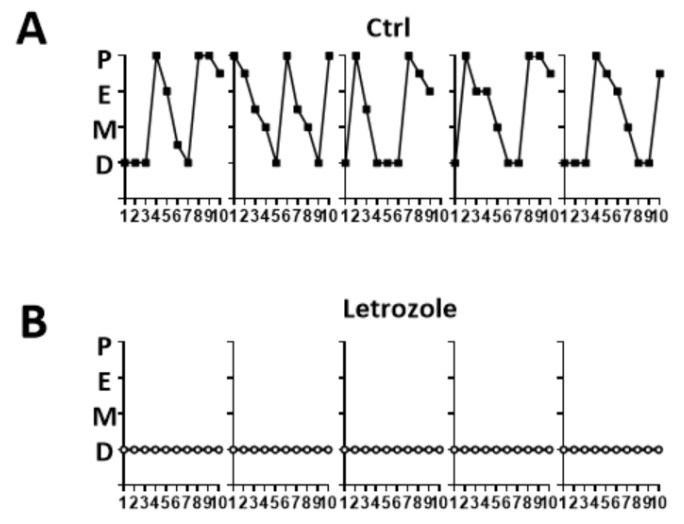
The estrous cycle pattern in the control group (**A**) and the letrozole-treated mice (**B**). The control mice showed a normal estrous cycle, whereas the estrous cycle of the letrozole-treated mice was arrested in the diestrus phase. P—proestrus; E—estrus; M—metestrus; D—diestrus.

**Figure 2 ijms-23-00134-f002:**
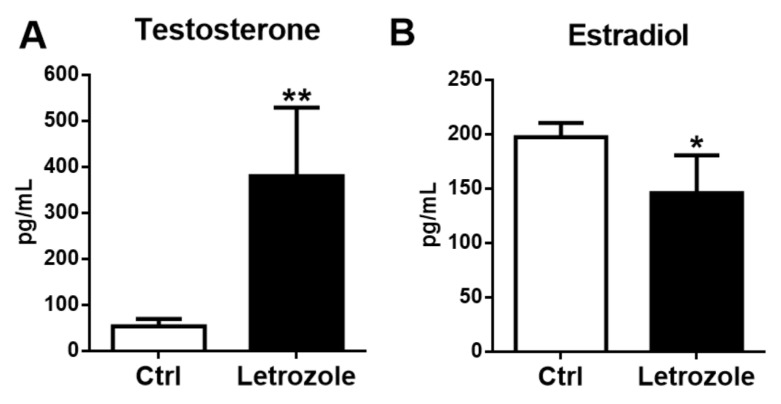
Plasma testosterone levels in the letrozole-treated mice were significantly higher than in the control mice (**A**), and estradiol levels were significantly lower in the letrozole-treated mice (**B**). Results are expressed as the mean ± SD. * *p* < 0.05; ** *p* < 0.01, compared with the control mice.

**Figure 3 ijms-23-00134-f003:**
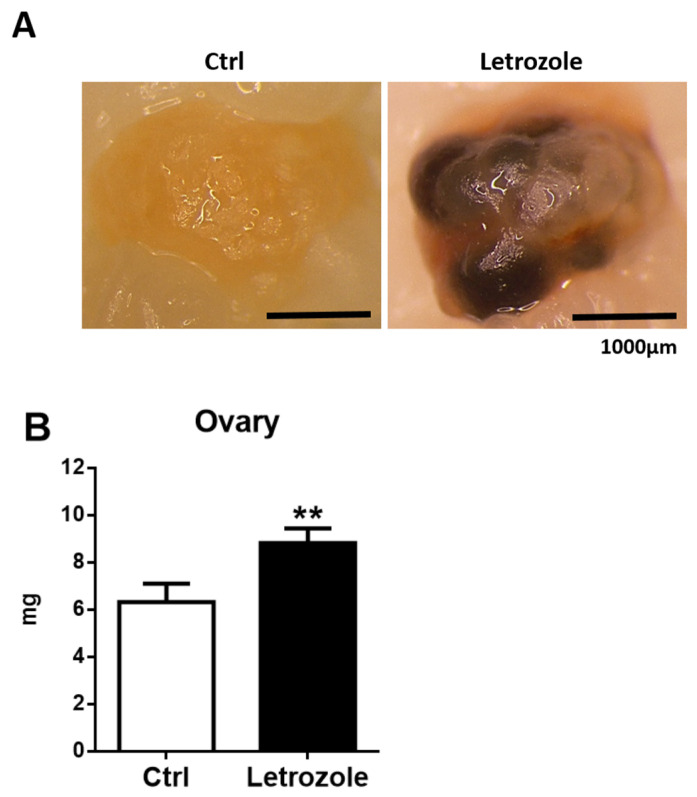
After 20 weeks of oral feeding with letrozole, the letrozole-treated mice showed some visible cysts on the surface of the ovaries, as shown in (**A**). The ovarian weight in the letrozole-treated mice is significantly higher than in control mice (**B**). The micromorphology of the ovaries was observed following hematoxylin and eosin (H&E) staining (**C**). The magnification is 40× and the scale bar = 250 μm. The ovary from a control mouse shows normal ovarian follicles (P—primary follicles; S—secondary follicles; A—antral follicles) and CL. The ovary from a letrozole-treated mouse shows cystic follicles (Cy) and the absence of the CL. ** *p* < 0.01, compared with the control mice.

**Figure 4 ijms-23-00134-f004:**
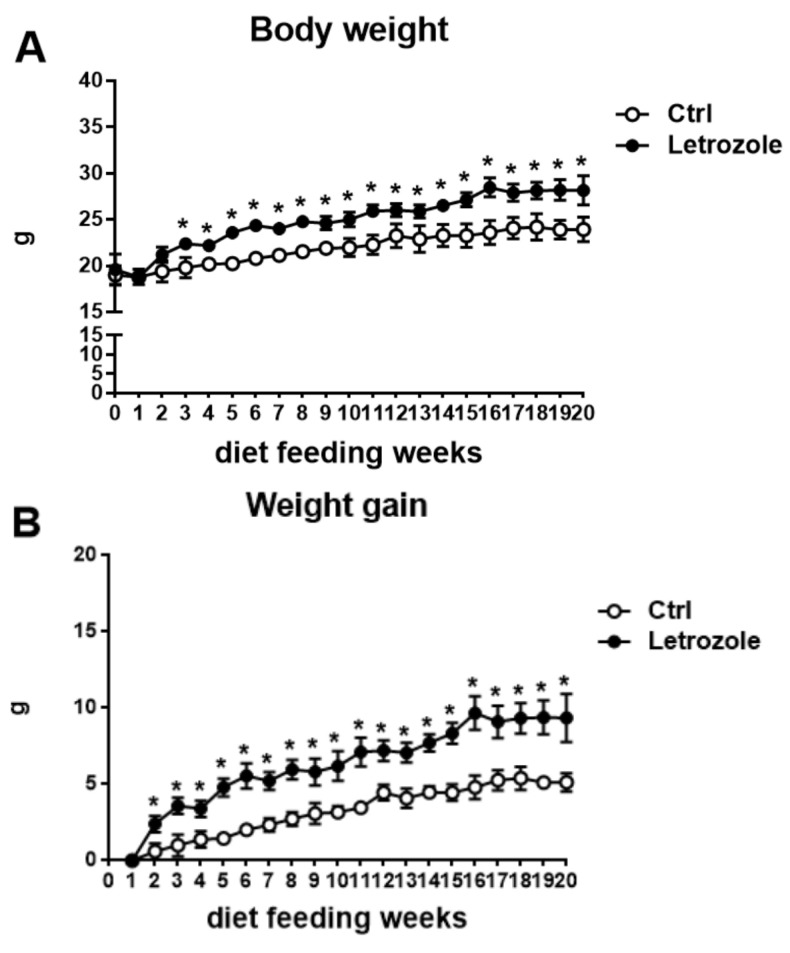
The body weight (**A**) and weight gain (**B**) of the letrozole-treated mice were significantly greater than in the control mice. The food intake showed no statistically significant difference between the two groups (**C**). Results are expressed as the mean ± SD. * *p* < 0.05, compared with the control mice.

**Figure 5 ijms-23-00134-f005:**
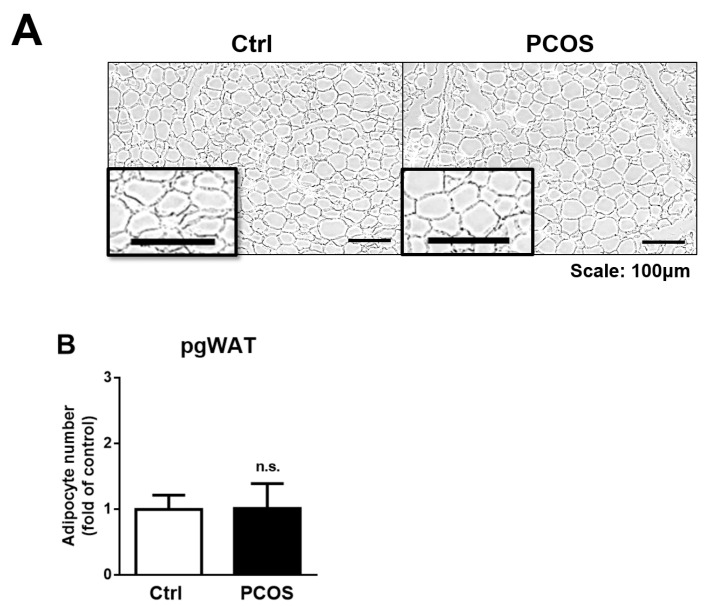
A histological image of the perigonadal white adipose tissue (pgWAT) is shown using H&E staining. Magnification = 400×. The virtual image of the adipose section (**A**) and the quantified result are both shown (**B**). Results are expressed as the mean ± SD. n.s.: not significant.

**Figure 6 ijms-23-00134-f006:**
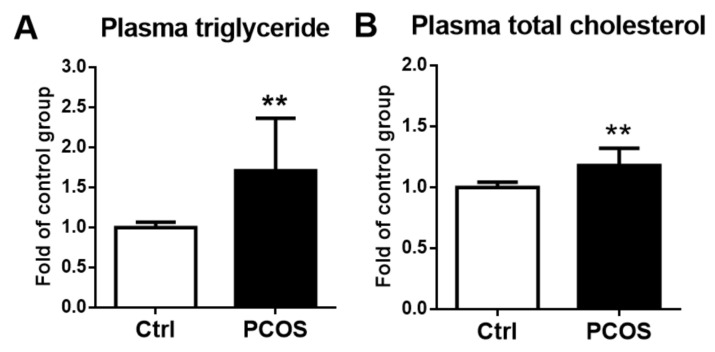
Plasma triglyceride (**A**) and total cholesterol (**B**) levels were significantly higher in the letrozole-treated mice. The results are expressed as the mean ± SD. ** *p* < 0.01, compared with the control mice.

**Figure 7 ijms-23-00134-f007:**
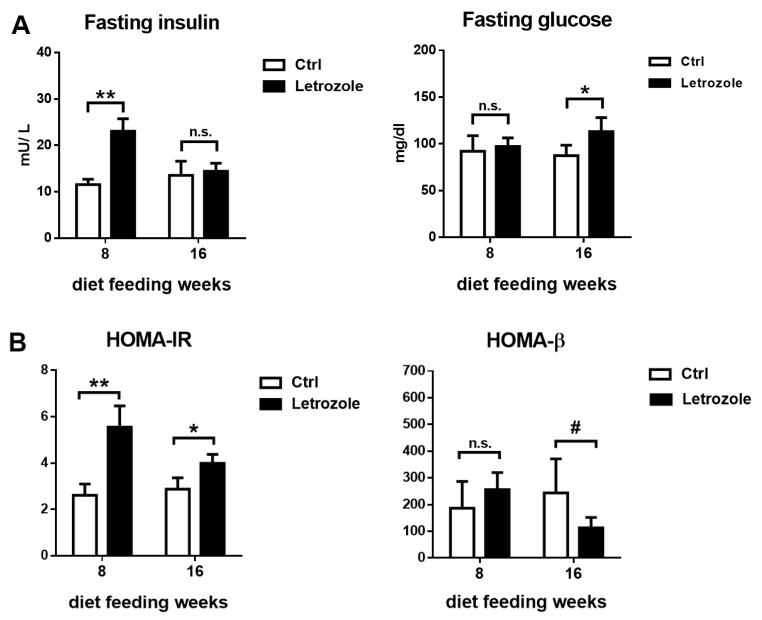
Comparing levels in fasting insulin, fasting glucose (**A**), HOMA-IR, and HOMA-β (**B**) levels from the letrozole-treated mice and the control groups at 8 and 16 weeks of oral feeding with letrozole. After 8 weeks of treatment with letrozole, the fasting plasma insulin levels were significantly higher in the letrozole-treated mice compared with the control mice. The HOMA-IR scores were significantly higher in the letrozole-treated mice after 8 and 16 weeks of treatment with letrozole. The results are expressed as the mean ± SD. ^#^ *p* < 0.1; * *p* < 0.05; ** *p* < 0.01, compared with the control mice. n.s.: not significant.

**Figure 8 ijms-23-00134-f008:**
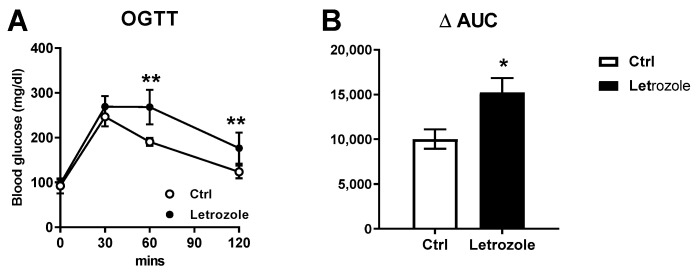
The oral glucose tolerance test (OGTT) was performed on the control and the letrozole-treated mice. The curves of plasma glucose (**A**) after glucose administration are shown. Using the ΔAUC calculation, the letrozole-treated mice showed significantly higher levels than the control mice (**B**). The results are expressed as the mean ± SD. * *p* < 0.05; ** *p* < 0.01, compared with the control mice.

**Figure 9 ijms-23-00134-f009:**
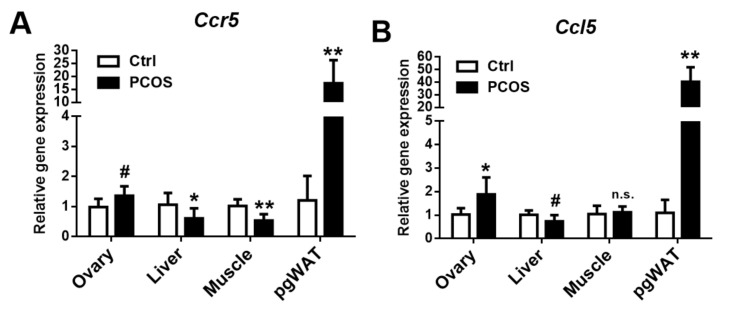
CCR5 (**A**) and CCL5 (**B**) expression in the ovary, liver, muscle, and perigonadal white adipose tissue (pgWAT) in the letrozole-treated and the control mice. The results are expressed as the mean ± SD. n.s.: not significant; ^#^ *p* < 0.1; * *p* < 0.05; ** *p* < 0.01, compared with the control mice.

**Figure 10 ijms-23-00134-f010:**
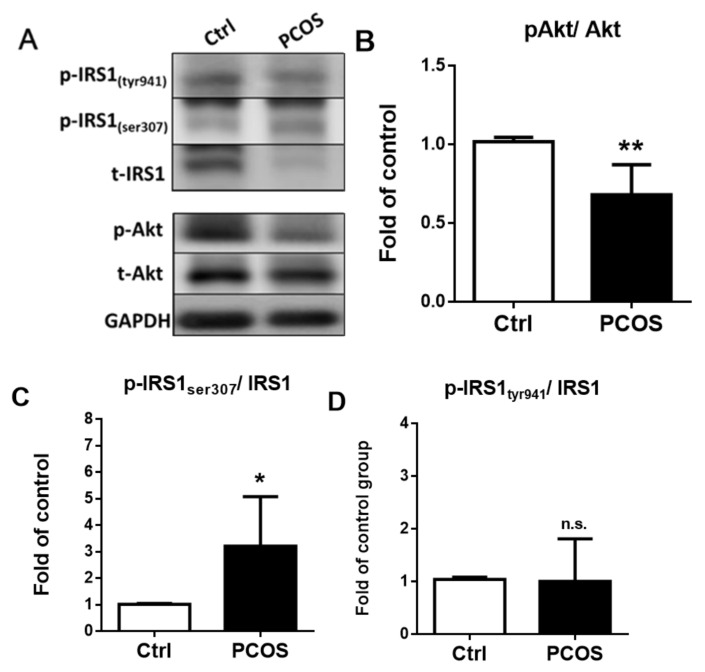
The insulin signal transduction pathway analysis of the perigonadal white adipose tissue (pgWAT) was performed in the letrozole-treated and control mice using Western blotting (**A**). Quantification analysis of relative Akt (**B**), IRS-1-Ser307 (**C**), and IRS-1-Tyr941 (**D**) phosphorylation in the pgWAT. Results are expressed as the mean ± SD. n.s.: not significant; * *p* < 0.05; ** *p* < 0.01, compared with the control mice.

**Figure 11 ijms-23-00134-f011:**
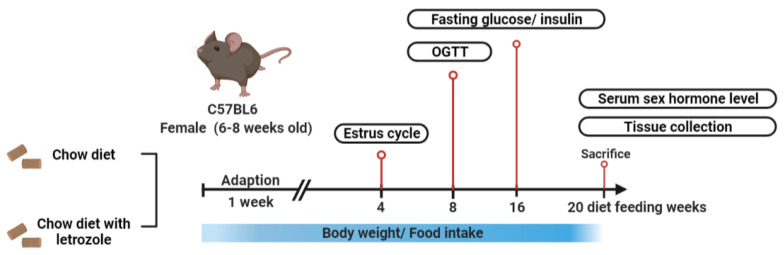
The procedure of the experiments.

## Data Availability

The data supporting the reported results of this study are included in the article.

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
