# Peer review of "Cysteine–Cysteine Motif Chemokine Receptor 5 Expression in Letrozole-Induced Polycystic Ovary Syndrome Mice"

_ijms, 2021, doi:10.3390/ijms23010134_

Round 1
Reviewer 1 Report
In the present study, Kok-Min Seow et al reported that treatment of female mice with letrozole for 8 weeks induced PCOS-like phenotypes. Moreover, the mice showed increased CCR5 and CCL5, which correlated with body weight, fasting insulin and HOMA-IR.
Letrozole has been widely used to induce PCOS-like phenotypes in rodents making the novelty of the present manuscript very limited. Indeed, the only new information from this study is the finding that letrozole increased CCR5 and CCL5 levels. However, it is not clear how does letrozole increase CCR5 and CCL5 protein expression. Does letrozole affect gene transcription or protein stability? Moreover, given that previous studies have rather reported letrozole-induced downregulation of these proteins, how do the authors explain their finding? Finally, the link between CCR5/CCL5 levels and insulin resistance is purely associative. Is insulin resistance a consequence of CCR5/CCL5 or the other way round?
The language of the manuscript needs to be improved.
Author Response
- Letrozole has been widely used to induce PCOS-like phenotypes in rodents making the novelty of the present manuscript very limited. Indeed, the only new information from this study is the finding that letrozole increased CCR5 and CCL5 levels.
Response: Thank you for your comments.
2. However, it is not clear how does letrozole increase CCR5 and CCL5 protein expression. Does letrozole affect gene transcription or protein stability?
Response: Thank you for your comment. Actually, what me know that letrozole only induced an animal model of PCOS with reproductive and metabolic abnormality. Letrozole did not alter the expression of CCR5. CCR5 over expression is related to the disease and insulin resistance of PCOS.
3. Moreover, given that previous studies have rather reported letrozole-induced downregulation of these proteins, how do the authors explain their finding?
Response: Thank you for your question. However, we could not find the related study that letrozole-induced PCOS downregulation of CCR5. We are the first study to find that letrozole-induced PCOS unregulated the expression of CCR5.
4.Finally, the link between CCR5/CCL5 levels and insulin resistance is purely associative. Is insulin resistance a consequence of CCR5/CCL5 or the other way round?
Response: Thank you for your comments. We found that CCR5 and CCL5 were significantly correlated with homeostasis model assessment of insulin resistance (HOMA-IR) in PCOS.
5. The language of the manuscript needs to be improved.
Response: Thank you for your suggestion. We have made a English revised of the manuscript.
Reviewer 2 Report
I read with great interest the manuscript, which falls within the aim of this Journal. In my honest opinion, the topic is interesting enough to attract the readers’ attention. Nevertheless, authors should clarify some points and improve the discussion, as suggested below.
Authors should consider the following recommendations:
- Manuscript should be further revised in order to correct some typos and improve style.
- Accumulating evidence suggests that one of the most important mechanisms of PCOS pathogenesis is the insulin-resistance. For this reason, the use of insulin-sensitizers, such as inositol isoforms, gained increasing attention due to their safety profile and effectiveness. Authors may better discuss this point, taking to account these recent articles: PMID: 26927948; PMID: 27579037.
Author Response
- Manuscript should be further revised in order to correct some typos and improve style.
Response: Thank you for your suggestion. We have English revised the manuscript.
2. Accumulating evidence suggests that one of the most important mechanisms of PCOS pathogenesis is the insulin-resistance. For this reason, the use of insulin-sensitizers, such as inositol isoforms, gained increasing attention due to their safety profile and effectiveness. Authors may better discuss this point, taking to account these recent articles: PMID: 26927948; PMID: 27579037.
Response: Thank you for your recommendation. We have discuss the role of inositol in discuss section (line 293-298).
Round 2
Reviewer 1 Report
The language of the revised manuscript is improved. However, the soundness of the finding remains low. Indeed, the authors did not make effort to assess my original comments.
A correlation between HOMA-IR and CCR5 and CCL5 does not provide any evidence on the link between these two parameters. Can the levels of CCR5 and CCL5 be also increased in other model of insulin resistance? This point is really crucial to understand whether CCR5 and CCL5 are the cause or consequence of insulin resistance in the letrozole-induced PCOS model. The authors should at least discuss this point.
The study that showed letrozole-induced downregulation was done in cell culture (doi: 10.1007/s12032-016-0779-z.). Although the focus of that study was not PCOS, the result could give a hind on the molecular mechanism linking letrozole and CCR5 expression and should be at least discussed.
Author Response
Thank you for your suggestion.
We agree that "A correlation between HOMA-IR and CCR5 and CCL5 does not provide any evidence on the link between these two parameter."
Other studies also found that CCR5 is associated with insulin resistance in ob/ob and HFD-induced mice.
Previous study by Kitade et al. (33) reported that CCR5 were upregulated in WAT of genetically (ob/ob) and high-fat diet (HFD)-induced obese (DIO) mice, and CCR5-/-mice were protected from insulin resistance, indicated that CCR5 may play a role in insulin resistance. Recent study also found that CCR5 knockout significantly attenuated glucose area under curve of OGTT and HOMA-IR in HFD-induced rats (34).
33. Kitade H, Sawamoto K, Nagashimada M, Inoue H, Yamamoto Y, Sai Y, et al. CCR5 plays a critical role in obesity-induced adipose tissue inflammation and insulin resistance by regulating both macrophage recruitment and M1/M2 status. Diabetes. 2012;61(7):1680-90.
34. Chan PC, Liao MT, Lu CH, Tian YF, Hsieh PS. Targeting inhibition of CCR5 on improving obesity-associated insulin resistance and impairment of pancreatic insulin secretion in high fat-fed rodent models. Eur J Pharmacol. 2021;891:173703.